# Competitive Learning in a Spiking Neural Network: Towards an Intelligent Pattern Classifier

**DOI:** 10.3390/s20020500

**Published:** 2020-01-16

**Authors:** Sergey A. Lobov, Andrey V. Chernyshov, Nadia P. Krilova, Maxim O. Shamshin, Victor B. Kazantsev

**Affiliations:** Neurotechnology Department, Lobachevsky State University of Nizhny Novgorod, 603950 Nizhny Novgorod, Russiak-nadezhda-k@yandex.ru (N.P.K.); nlhr@yandex.ru (M.O.S.); kazantsev@neuro.nnov.ru (V.B.K.)

**Keywords:** EMG interface, STDP, pair-based STDP, triplet-based STDP, temporal coding, rate coding, synaptic competition, neural competition, lateral inhibition

## Abstract

One of the modern trends in the design of human–machine interfaces (HMI) is to involve the so called spiking neuron networks (SNNs) in signal processing. The SNNs can be trained by simple and efficient biologically inspired algorithms. In particular, we have shown that sensory neurons in the input layer of SNNs can simultaneously encode the input signal based both on the spiking frequency rate and on varying the latency in generating spikes. In the case of such mixed temporal-rate coding, the SNN should implement learning working properly for both types of coding. Based on this, we investigate how a single neuron can be trained with pure rate and temporal patterns, and then build a universal SNN that is trained using mixed coding. In particular, we study Hebbian and competitive learning in SNN in the context of temporal and rate coding problems. We show that the use of Hebbian learning through pair-based and triplet-based spike timing-dependent plasticity (STDP) rule is accomplishable for temporal coding, but not for rate coding. Synaptic competition inducing depression of poorly used synapses is required to ensure a neural selectivity in the rate coding. This kind of competition can be implemented by the so-called forgetting function that is dependent on neuron activity. We show that coherent use of the triplet-based STDP and synaptic competition with the forgetting function is sufficient for the rate coding. Next, we propose a SNN capable of classifying electromyographical (EMG) patterns using an unsupervised learning procedure. The neuron competition achieved via lateral inhibition ensures the “winner takes all” principle among classifier neurons. The SNN also provides gradual output response dependent on muscular contraction strength. Furthermore, we modify the SNN to implement a supervised learning method based on stimulation of the target classifier neuron synchronously with the network input. In a problem of discrimination of three EMG patterns, the SNN with supervised learning shows median accuracy 99.5% that is close to the result demonstrated by multi-layer perceptron learned by back propagation of an error algorithm.

## 1. Introduction

Nowadays, artificial neural networks (ANN) are widely used in practical applications. One of the important applications is the use of ANN in the human–machine interface (HMI), in particular in the electromyographical (EMG) interface. Several strategies are used to solve the problem of control of external (“additive”) devices using EMG signals. Conventional techniques are based on one-channel recordings and limited to either trigger control based on detecting a threshold signal, or proportional control in the case of continuous monitoring of some discriminating feature extracted from the EMG signal. Multichannel recording significantly expands control capabilities, and entirely new signal-processing techniques are used, such as classification of EMG patterns and multichannel regression [1,2]. EMG classification (accordingly, movement recognition) can be combined with command control and, hence, can be used in the case when a control device is equipped with an autonomous, local low-level control system capable of implementing some macro commands. ANN-based technics are used in a vast range of EMG classification tasks (see, e.g., [3,4,5]). In turn, regression allows one to estimate muscle effort strength by its EMG signal and, hence, can be used for proportional (gradual) control, e.g., for reconstruction of torque value of some joints [6]. In addition to other mathematical tools, ANNs were also successfully applied to the regression problem, including multichannel registration [7,8,9].

ANNs based on artificial neurons are widely used in approximation, classification and clustering problems. The artificial neuron, originally proposed by McCulloch and Pitts [10] represents a weighted input adder with an output activation function. Such simplification of a living neuron permits the development of a variety of effective learning procedures (see e.g., [11,12]). However, artificial neurons used in ANNs still cannot solve many neurocomputational “dynamical” tasks based on temporal encoding and synchronization effects.

Spiking neural networks (SNN) are a relatively new paradigm for neural computations. A spiking neuron represents a dynamical system, where a spike fires when a neuron membrane potential exceeds a certain excitation threshold. Moreover, this excitation may be activity dependent, e.g., be determined by previous activity of a neuron [13]. It is believed that the SNN-based computations have great potential [14,15] and, theoretically, can reach enormous computational efficacy like real brain circuits.

A spiking neuron with synaptic plasticity is considered as an ideal candidate for the role of the basic element of upcoming neuromorphic systems based on memristive devices (see i.e., [16,17]). Their key ability to synchronize [18,19] can be used in artificial, additive devices and prostheses [20,21]. The neural synchronization in such systems can involve both living and artificial neurons. There are reports showing the first experimental implementations of hybrid networks consisting of living neuronal cells and artificial spiking neurons [22,23]. Thus, the new generation of HMI can be implemented entirely by neural networks where neurons of the brain interact with their artificial counterparts that work as part of prostheses or external devices.

Neural network functions depend on learning defined basically by tuning weights of couplings between the network units. There are several learning rules used in ANNs including Hebbian learning. However, Hebb did not give any mathematical equations for his idea about the potentiation of interacting neurons whose activity correlates [24] the learning rule can be written as [12,25]:(1)$$\Delta w_{ij}=\eta x_{j}y_{i}$$
where $\Delta w_{ij}$ is the change of coupling from neuron *j* to neuron *i*, η is learning rate, $x_{j}$ is output activity of neuron *j* (input signal for neuron *i*), $y_{i}$ is output activity of neuron *i*. Equation (1) cannot be used in such form because it may lead to unlimited increase of the weights. This problem can be solved, in particular, by introducing the *forgetting function* that depends on the output activity of the neuron and on the weight of the input connection [26]:(2)$$\Delta w_{ij}=\eta x_{j}y_{i}-F\left(y_{i}\right)w_{ij}$$

Taking into account some restrictions [27], one can transform Equation (2) to the rule of competitive learning widely used in ANNs to implement unsupervised learning:(3)$$\Delta w_{ij}=\{\begin{array}{c} \eta (x_{j}-w_{ij}),\;\mathrm{if}\;\mathrm{neuron}\;i\;\mathrm{wins}\;\mathrm{competition}\;\left(y_{i}=1\right)\\ 0,\;\mathrm{if}\;\mathrm{neuron}\;i\;l\mathrm{oses}\;\mathrm{competition}\;\left(y_{i}=0\right) \end{array}$$

This is the so called “winner takes all” rule meaning that only the neuron that has maximum output response to the input pattern can be trained.

In contrast with ANN, in SNN one can use an experimentally confirmed [28,29,30] algorithm of Hebbian learning in the form of spike timing-dependent plasticity (STDP). The STDP potentiates coupling between two neurons if a postsynaptic neuron generates a spike after a presynaptic one and depresses it otherwise [31]. It is important to note that this type of plasticity includes elements of synaptic competition, which makes «success» of the synapses dependent on the time of spikes transmitted through it [32].

Earlier we proposed to use layer of spiking neurons as a feature extractor for EMG. A signal from SNN was transmitted to ANN that classified EMG patterns corresponding to different hand gestures [21]. The aim of the current study is to develop an intelligent classification system based entirely on SNN. To do this we first explore the possibility of rate and temporal coding by one neuron and then define a minimal set of basic learning rules to ensure a selective SNN’s response. Then, we implement the studied principles in a concrete SNN classifying the EMG patterns. The developed SNN can be used in upcoming neuromorphic systems as a core implementing HMI.

## 2. Models and Methods

For a single spiking neuron we employed dynamical system proposed by Eugene Izhikevich [33]. The neuron’s driving current is given by:(4)$$I\left(t\right)=\xi \left(t\right)+I_{syn}\left(t\right)+I_{stml}\left(t\right)$$
where ξ(t) is an uncorrelated zero-mean white Gaussian noise with variance D, $I_{syn}\left(t\right)$ is the synaptic current, and $I_{stml}\left(t\right)$ is the external stimulus. The synaptic current represents the weighted sum of all synaptic inputs to the neuron:(5)$$I_{syn}\left(t\right)=\sum_{j}g_{j}w_{j}\left(t\right)y_{j}\left(t\right)$$
where the sum is taken over all presynaptic neurons, $w_{j}$ is the strength of the synaptic coupling directed from neuron j, $g_{j}$ is the scaling factor equal either to 2 or to −2 for excitatory and inhibitory neurons, respectively, and $y_{j}\left(t\right)$ describes the amount of neurotransmitters released by presynaptic neuron j: (6)$$\frac{dy_{j}}{dt}=-\frac{y_{j}}{\tau }+\sum_{t_{j}^{sp}}\delta \left(t-t_{j}^{sp}\right)$$
where τ = 100 ms is the decay time of synaptic output [31].

We implemented the STDP model using local variables or traces [31]. The weight increase corresponding to long-term potentiation (LTP) occurs when a postsynaptic neuron fires a spike and it is proportional to presynaptic trace $y_{j}^{1}\left(t\right)$:(7)$${\frac{dw_{ij}}{dt}}_{+}=F_{+}\left(w_{ij}\right)y_{j}^{1}\left(t\right)\sum_{t_{i}^{sp}}\delta \left(t-t_{i}^{sp}\right)$$

The weight decrease corresponding to long-term depression (LTD) occurs when a presynaptic neuron fires a spike and it is proportional to a postsynaptic trace $y_{i}^{1}\left(t\right)$:(8)$${\frac{dw_{ij}}{dt}}_{-}=-F_{-}\left(w_{ij}\right)y_{i}^{1}\left(t\right)\sum_{t_{j}^{sp}}\delta \left(t-t_{j}^{sp}\right)$$

For the weight updating, we use the multiplicative rule [34]:(9)$$F_{+}\left(w_{ij}\right)=\lambda \left(1-w_{ij}\right),\;F_{-}\left(w_{ij}\right)=\lambda \alpha w_{ij}$$

For the rate coding we also used the triplet-based STDP characterized by frequency dependence [35]. Unlike the pair-based rule, the triplet-based rule uses two local variables—fast and slow with different decaying times $\tau_{1}$ and $\tau_{2}$, and the dynamics of these variables can be also described by Equation (6). In the minimal triplet model [35] the LTD is calculated by Equation (8), but in the LTP the increase of weight is proportional not only to the fast presynaptic trace, $y_{j}^{1}\left(t\right)$, but also to the slow postsynaptic trace, $y_{i}^{2}\left(t\right)$, as follows:(10)$${\frac{dw_{ij}}{dt}}_{+}=F_{+}\left(w_{ij}\right)y_{j}^{1}\left(t\right)y_{i}^{2}\left(t\right)\delta \left(t-t_{i}^{sp}\right)$$

We used the following parameter values: *λ* = 0.001, *α* = 1, $\tau_{1}$ = 10 ms, $\tau_{2}$ = 100 ms (corresponding to minimal triplet model in [35]).

First, let us consider temporal and rate coding for single neuron. The scheme of the network is illustrated in Figure 1. Each of 10 presynaptic neurons encodes time or frequency of spikes in the repeating input patterns affecting the postsynaptic neuron during learning. In temporal coding (Figure 1A), stimulation pattern contained definite sequence of pulses S_1_–S_10_ with the inter-pulse interval ∆*t* taken here values of 1, 2, 5, 10 and 20 ms in different simulations. The frequency of such stimulus applications was 1 Hz. In the rate coding (Figure 1B), we tuned stimulation parameters so that the presynaptic neurons fired spike trains with average frequencies 0.1, 0.2, 0.5, 1, 2, 3, 6, 12, 25 and 50 Hz. In our simulations, the learning protocol lasted 1000 s of model time.

We used familiar (e.g., learned before) and unknown patterns to estimate the result of learning in both coding schemes. In the temporal coding, we took the first/last half of the temporal pattern as a familiar/unknown pattern, respectively. In the rate coding, in order to generate the unknown pattern, we reversed the learned pattern so that the first and the last presynaptic neurons had a spiking rate 50 Hz and 0.1 Hz, respectively.

For experimental purposes, we recruited 8 healthy volunteers of either sex from 18 to 44 years old. The study complied with the Helsinki declaration adopted in June 1964 (Helsinki, Finland) and revised in October 2000 (Edinburg, Scotland). The Ethics Committee of the Lobachevsky State University of Nizhny Novgorod approved the experimental procedure (protocol No. 35 from 5 September 2019). All participants gave their written consent.

Registration of the EMG signals was accomplished with the use of 8-channel bracelet MYO Thalmic Labs, which was located on subject’s forearm. During SNN learning, each subject in a standing position alternately flexed and extended his/her wrist for one minute. Meanwhile each gesture—rest, flexion and extension of the hand—lasted about 3 s. SNN learning was performed online directly at the time of EMG registration. However, we measured the accuracy of classifying EMG patterns by offline records. It was equal to the ratio of the spike rate of the classifier neuron excited by the presentation of “its own pattern” to the sums of spike rates of all three classifiers.

To estimate the gradual character of SNN activity, we asked the subjects to flex and extend their wrist with four different degrees of effort, determined by the different degrees of deviation of the palm from the center position. Each pattern was 10 s long and was sent to the input of trained SNN. The muscle effort strength was estimated indirectly through mean absolute value (MAV) of the EMG signal, which was averaged on the whole time interval over all EMG channels.

## 3. Results

### 3.1. Spiking Neurons as Electromyographical (EMG) Features Extractors

One of important information features of the EMG signal is its amplitude. Earlier we proposed method to extract this feature using spiking neurons [21]. In particular, a “sensory” neuron receives from a virtual stimulator a signal in the form of EMG-associated current:(11)$$I_{stml}\left(t\right)=k\cdot EMG\left(t\right)$$
where *EMG*(*t*) denotes recorded EMG signal and *k* is the scaling coefficient (we use *k* = 2 × 10^6^ as in [21]).

Figure 2 shows an example of neural activity of two sensory neurons receiving inputs from electrodes located on extensors during wrist extension. Both registered muscles take part in the current movement, however, signals from them have different amplitudes due to the anatomical properties of these muscles and/or to the localization of the electrodes (Figure 2, top panel). Both input signals lead to increasing spiking frequency rate of corresponded sensory neurons (Figure 2, S3, S5) and the EMG channel with higher amplitude evokes faster spiking (Figure 2, red line). Thus, the spiking neurons perform rate coding. The spiking rate depends on the amplitude of the EMG signal, which, in turns, corresponds to muscle strength. 

However, there are different latencies of spiking response to EMG signal of various amplitude. A sensory neuron receiving the signal of lower amplitude (Figure 2, blue line) begins to respond to it much later compared with stronger stimuli (Figure 2, red line). Thus, a spiking neuron simultaneously encodes the input signal based both on the spiking rate and on latencies of the first response spike. In the case of such temporal-rate coding, the SNN should implement learning mechanisms worked properly for both types of coding. Based on this, we first studied the training of a single neuron with a pure rate and temporal patterns, and then built a universal SNN that is trained using mixed coding.

### 3.2. Learning and Selective Response of a Single Neuron

In temporal coding learning neuron receives information as a sequence of spikes from different presynaptic neurons. Consequently, we expect to obtain weight distribution depending on spike timing within the training pattern and (in the protocol used) on the rank of spiking. Indeed, in both cases of the STDP (pair- and triplet-based) after repeating stimulation, we find correlations between weights and spike timing (Figure 3A, solid lines). This effect can be explained by the presence of a refractory period in spiking neurons. After firing a spike, a postsynaptic neuron receives presynaptic spikes in the after-spike hyperpolarization period reproduced by the Izhikevich model. Consequently, the neuron cannot respond and corresponding couplings become depressed. Time intervals between spikes varied from 1 to 10 ms in simulations, then, the time of the pattern presentation was varied from 10 to 100 ms. In the case of shorter time intervals (<5 ms), the weights of the first couplings become potentiated, while the rest become depressed. In the case of increased interval, the neuron have enough time to recover its sensitivity within the pattern, which leads to alternating couplings with large and small weights (Figure 3A, dashed lines).

Let us consider the selective response of the neuron to a familiar pattern as a criterion for success learning. In the case of short interspike intervals and the weight dependence on the rank of spiking (Figure 3A, solid lines), the postsynaptic neuron shows high/no response activity to the familiar/unknown patterns, respectively (Figure 3B, 4 ms). In the case of big intervals and alternating weights (Figure 3A, dashed lines), the neuron is almost unable to discriminate the patterns (Figure 3B, 10 ms). The pair- and triplet-based STDP rules have similar weight distributions and selectivity in all studied cases (Figure 3).

Thus, a single neuron can potentially be selective to the rank of spiking only at the beginning of the temporal pattern. This effect was described earlier [36]; on its basis, STDP-driven latency coding can be implemented, in which synapses that transmit spikes faster decrease their latency [37]. In general, the SNN needs to implement neural competition and axonal delays for encoding complex and long temporal patterns [38]. The sensitivity of STDP-driven neuron to the beginning of temporal pattern can lead to spatial heterogeneity of monolayer a SNN under local repeating stimulation. Each neuron in such a SNN after “learning” has potentiated its input connections from the stimulation side and depressed ones from the opposite direction. At the network scale as a result the centrifugal (relative to the stimulation site) couplings are potentiated and network responses become synchronized to stimuli [39,40].

Attempts to implement rate coding based only on STDP failed in our experiments. There are no expected relations between weights distribution and frequency rate of the stimuli (Figure 4A, “STDP” and “tSTDP”). Accordingly, no neural selectivity was observed (Figure 4B, “STDP” and “tSTDP”). This happened because of the STDP events (close pairs and triplets of spikes) do not depend on the presynaptic frequency rate. Constant stimulation with the rate pattern leads to fluctuations of refractory durations of the postsynaptic neuron. During the excitable state of this neuron the incoming spikes make it fired regardless of their frequency rate. It corresponds to the presynaptic- postsynaptic (“pre-post”) spike sequence and STDP potentiates couplings. Other spikes of all frequency rates arrive at the refractory stage. It corresponds to the “post-pre” sequence and STDP depress coupling. As a result, all weights become averaged regardless of the frequency rate.

The LTP part of triplet-based STDP for spiking neurons (Equation (10)) is most consistent with the Hebbian learning for artificial neurons (Equation (1)). Accordingly, they have the common drawback—unlimited weight growth. More precisely, when applying the multiplicative rule (Equation (9)), the weight is limited to 1. The problem is that the triplet-based STDP depends on the averaged frequency of the postsynaptic neuron only and, regardless of the rate of presynaptic spikes, it potentiates all incoming couplings. In other words, there is the lack of synaptic selectivity and as a result, the neuron cannot discriminate patterns (Figure 4B, tSTDP).

The synaptic competition can be a possible solution of this problem. Similar to the ANNs (Equation (2)), we introduce *forgetting function* for incoming synapses, which is proportional to neuronal activity:(12)$$\frac{dw_{ij}}{dt}=-\frac{w_{ij}y_{i}}{\tau_{f}}$$
where $\tau_{f}$ is decay time of weights, $y_{i}$ describes averaged activity of postsynaptic neuron *i* described Equation (6) with different decay time of synaptic trace *τ**_o_*.

Using the triplet-based STDP combined with the forgetting function (parameters $\tau_{f}$ = 10 ms, *τ_o_* = 100 ms) one can gain explicit dependence of weights on the presynaptic spike rate (Figure 4A, tSTDP + F). Note, that the relation is strictly sigmoid. Selectivity testing shows that postsynaptic neuron activity during exposition of familiar pattern is considerably higher than in the case of unknown pattern (Figure 4B, tSTDP + F).

### 3.3. EMG Patterns Classification Problem as an Example of Unsupervised Learning in Spiking Neuron Networks (SNN)

Next, we tested the new learning rule of the triplet-based STDP with synaptic forgetting to design a SNN capable of classifying EMG patterns. Unlike the case of individual neurons, training the whole network should provide recognition of several patterns. Therefore, the structure of the SNN (number of neurons and neural layers, topology of neural connections, etc.) should be built specifically to solve this task. Proposed SNN (Figure 4A) consists of two layers with “sensory” and “classifying” functions (“S” and “C”, respectively, in Figure 5). In turn, each layer includes excitatory and inhibitory neurons.

Inhibitory neurons (Figure 5, marked blue) in the input layer are necessary for lateral inhibition, which significantly improves the quality of further recognition of EMG patterns by contrasting the signal [21]. In order to identify the muscle rest patterns we include one additional neuron in the input layer, which fires spikes when the other input neurons are silent. For this propose we use large individual noise (*D* = 70) for this neuron and strong incoming couplings from inhibitory neurons.

The output network layer consists of three excitatory neurons that classify EMG signals after learning (Figure 5A, “classifiers”) and three inhibitory neurons that provide lateral inhibition. In this case, lateral inhibition plays a key role in learning: when one of the neurons-classifiers is active, the other output neurons are inhibited. As the learning rule (triplet-based STDP, synaptic forgetting) works only while postsynaptic neuron is active, only one neuron can be trained at a time. Thus, lateral inhibition implements the “winner takes all” principle which is widely used in traditional ANNs implementing self-organizing maps (SOM) proposed by Kohonen [27]. As a result of learning, the coupling strengths between input and output layers change providing a selective response to different EMG patterns (Figure 5B–D, Appendix A). As the proposed SNN is based on unsupervised learning, it is unpredictable to say which neuron will respond to a particular pattern. Therefore, if we use SNN as a classifier we need to assign class labels to output neurons after learning.

During learning procedure of about 1 min, raw EMG signals were sent online to the input layer of SNN while a subject flexed and extended his/her wrist (Appendix A). Figure 6 illustrates typical EMG signals and responses of trained classifier neurons. Note that the neurons make errors predominantly when EMG patterns (correspondently hand movements) are changed.

With selected SNN parameters median accuracy for the eight subjects was 91% (Q_1_ = 85%, Q_3_ = 95%) which was lower than the 100% accuracy demonstrated by multi-layer perceptron with a back propagation algorithm applied to the same problem. But it would be more correct to compare the proposed SNN with Kohonen’s SOM, where competitive learning is performed in corresponding ANN [27]. Earlier we showed, that a SOM-based classifier demonstrated median accuracy 87 % for five EMG patterns [41]. In the current study median accuracy of SOM for eight participants was 88% (Q_1_ = 82%, Q_3_ = 89%) for the three motions.

Figure 7A shows the distribution of the normalized amplitude of the EMG signal averaged over all subjects when performing wrist flexion and extension. This profile corresponds to the distribution of weight coefficients of two trained classifier-neurons that can be selectively excited when these movements are performed (Figure 7B). Thus, the combination in the SNN of the triplet-based STDP, synaptic forgetting and lateral inhibition leads to the formation of a distribution of weights similar to the distribution of the amplitude feature of the input signal. Thus, the proposed complex learning rule for our SNN works quite similar to the competitive learning implemented in an ANN (Equation (3)).

In addition, the proposed SNN shows a gradual nature of the response depending on the amplitude of the signal. In particular, the dependence of the spike rate of classifier neurons on the amplitude of the EMG signal is linear (Figure 7C). Considering that the amplitude of the EMG, in turn, is also linearly proportional to the effort developed by the muscles [42], it can be concluded that classifier neurons not only recognize the movement performed by the subject, but also encode the degree of muscle strength involved in such movements.

### 3.4. SNN Supervised Learning 

Next, we developed supervised SNN learning. In contrast with unsupervised learning, we now stimulate target neurons during pattern presentation to the network input. Technically, in our neuro-simulator application at the time moment of the EMG pattern presentation we connect the virtual stimulation electrode that generates high-frequency activity (40 Hz) to one of the classifier neuron (Appendix A). This leads to excitation of the target neuron and inhibition of the other classifier neurons. As a result only one target neuron “associates itself” with the presented EMG pattern. Next, this “supervised stimulation” was applied to another target classifier neuron during another EMG pattern presentation to the network input. Note that there is no need to deactivate learning in time intervals between stimuli—during this time the triplet-based STDP and synaptic forgetting are working but not erasing previous results. Earlier, similar mechanism called Pavlov’s principle was proposed as an analog of backpropagation error method in SNN [43]. In our case, we also generate SNN feedback via additional stimulation labeling the neurons that planned to be trained at a time.

After such online procedure of supervised learning median accuracy of SNN was 99.5% (Q_1_ = 99.4%, Q_3_ = 99.8%). Note that these results are much closer to 100% accuracy of the multi-layer perceptron than in the case of SNN unsupervised learning.

## 4. Discussion

In summary, we have shown the possibility of implementing competitive learning in spiking neurons in the context of temporal and rate coding. We have demonstrated that for such learning the following three major mechanisms should be employed together, including:(i)Hebbian learning (in the current work, through triplet-based STDP);(ii)synaptic competition or competition of inputs (in the current work, through synaptic forgetting); and(iii)neural competition or competition of outputs (in the current work, through lateral inhibition).

The use of Hebbian learning in the form of pair- or triplet-based STDP is sufficient for temporal coding. In this case, the neurons are sensitive only to spikes in the beginning of the input pattern. A neural network with neural competition (lateral inhibition) and axonal delays is required for encoding of complex and long-term patterns [38].

However, Hebbian learning only is not sufficient to implement rate coding. In this case, to enrich the selectivity, one should employ synaptic competition which ensures depression of less used synapses. We have implemented this type of competition by introducing the *forgetting function* for incoming synapses proportional to the activity of the postsynaptic neuron. Obviously, synaptic competition can be implemented in other ways, for example, using homeostatic plasticity [44,45]. Hence, by combining Hebbian learning with synaptic competition, both temporal and rate coding can be achieved. Moreover, learning-driven weights rearrangement determines by type of coding, rather than a priori specified network topology [44].

Note that here we do not study carefully the quality of the selectivity achieved by training one neuron. In the case of both temporal and rate coding, to test selectivity we use a pattern that is very different from that learned. Note also, that recently the concept of a multidimensional brain has been proposed for ANNs, according to which the neuron selectivity increases non-linearly with increasing dimension (number) of synaptic inputs. In particular, when certain (rather general) conditions are met for an artificial neuron the theoretically achievable selectivity can approach 100% with a number of synapses of more than 20 [46,47]. When using 10 synapses, as in the current work (learning one neuron), the theoretical selectivity is about 50%, which means that in the space of input patterns even a perfectly learned neuron will classify about half of the patterns as familiar. Obviously, in the case of spiking neurons, we can expect similar dimensional dependence and its study could be the subject of our future work.

Neural competition is necessary for selective SNN learning: not all output neurons should respond to a particular pattern, but only a part. As a result, different neurons or neural groups will acquire an affinity for different input patterns. In our SNN, for this purpose we introduced lateral inhibition permitted to implement the “winner takes all” principle. Earlier, we also used lateral inhibition in processing the EMG signal enhancing the contrast [21].

Unsupervised learned SNN cannot compete with a multilayer perceptron in the classification accuracy. Nevertheless, even in its simple form it has several advantages based on the analogous signaling of spiking neurons. In particular, the SNN can provide gradual response depending on input signal amplitude and the low lag of response to the change of input pattern. Note also, that earlier we proposed some improvements of EMG control based on ANNs, in particular combined command-proportional EMG control [42] and optimizing response speed [48]. However, these extensions of basic ANN functions required special configurations of EMG interfaces and the use of external non-ANN algorithms.

Finally, we proposed a simple implementation of supervised learning in SNN. The single-layer architecture is so far more similar to the classical Rosenblatt’s perceptron than to the multi-layer ANN, trained by the error back-propagation algorithm. Nevertheless, in the problem of discrimination of three EMG patterns the supervised learned SNN shows accuracy close to the result demonstrated by the multi-layer perceptron learned by back propagation of the error algorithm. It is shown that SNN learning based on error correction can act similarly to the back propagation in the perceptron (see [49,50]). In our model, SNN implements biologically plausible associative learning by the associations of certain input patterns with the activity of certain output neurons. Thinking about further developments, a design of multi-layer SNN will be proposed in which the input and hidden layers provide unsupervised competitive learning, while the output layer can be trained using the proposed “supervised stimulation”.

## Figures and Tables

**Figure 1 sensors-20-00500-f001:**
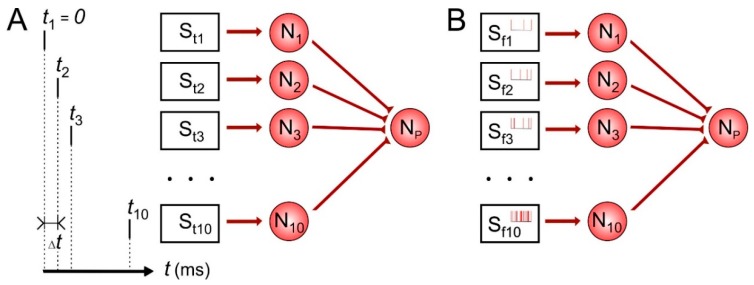
Temporal (**A**) and rate (**B**) coding scheme for single neuron. S_t1–t10_, S_f1–f10_—stimulators with temporal and frequency rate parameters correspondently, N_1–10_—presynaptic neurons, Np—postsynaptic neuron, *t_j_*—stimulating pulse time for neuron *N_j_*. ∆*t* = *t_j_*_+1_ − *t_j_*—time between pulses.

**Figure 2 sensors-20-00500-f002:**
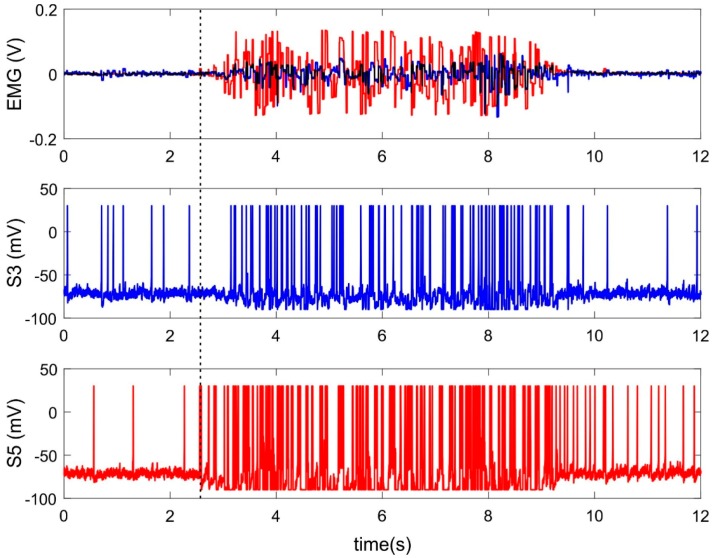
Features extraction by spiking neurons. Top panel: example of two electromyographical (EMG) channels recording muscle activity of *m. extensor carpi radialis* (blue) and *m. extensor carpi ulnaris* (red) during wrist extension. S3, S5—activity (the membrane potential) of sensory neurons receiving input from the 3rd and 5th electrodes of the MYO Thalmic bracelet registering EMG from these muscles. The dashed line indicates the start of movement.

**Figure 3 sensors-20-00500-f003:**
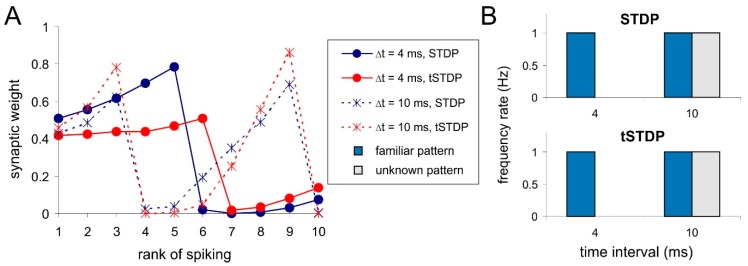
Temporal coding results of simulations with different inter-spike intervals using pair-based rule (spike timing-dependent plasticity, STDP) and triplet-based rule (tSTDP): (**A**) Synaptic weights vs. the rank of spiking on corresponding synapses; (**B**) The response of the postsynaptic neuron to familiar and unknown pattern.

**Figure 4 sensors-20-00500-f004:**
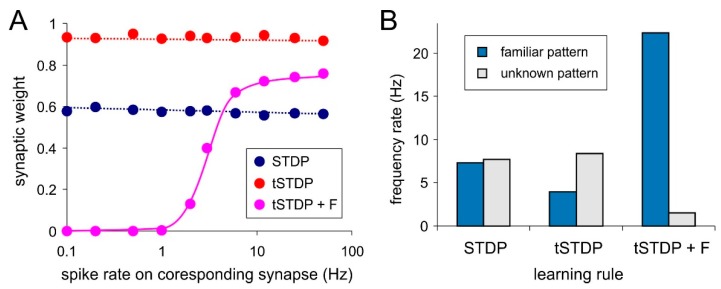
Rate coding results with pair-based rule (STDP) triplet-based rule (tSTDP) and triplet-based rule with forgetting function (tSTDP + F): (**A**) Synaptic weights vs. spike rate on corresponding synapse; (**B**) The response of the postsynaptic neuron to familiar and unknown pattern.

**Figure 5 sensors-20-00500-f005:**
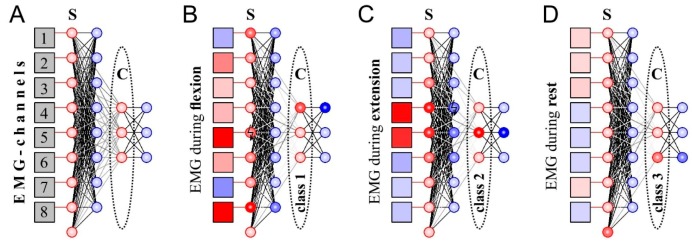
The scheme of spiking neuron networks (SNN) classifying EMG-patterns: (**A**) General topology of SNN; (**B**–**D**) The responses of SNN to EMG patterns during wrist flexion/extension/rest. Squares—EMG-associated virtual electrodes, red/blue circles—excitatory/inhibitory neurons. The color saturation of red/blue is proportional to the value of the positive/negative output signal of the corresponding circuit element at the moment. S—sensory neurons, C—classifying neurons.

**Figure 6 sensors-20-00500-f006:**
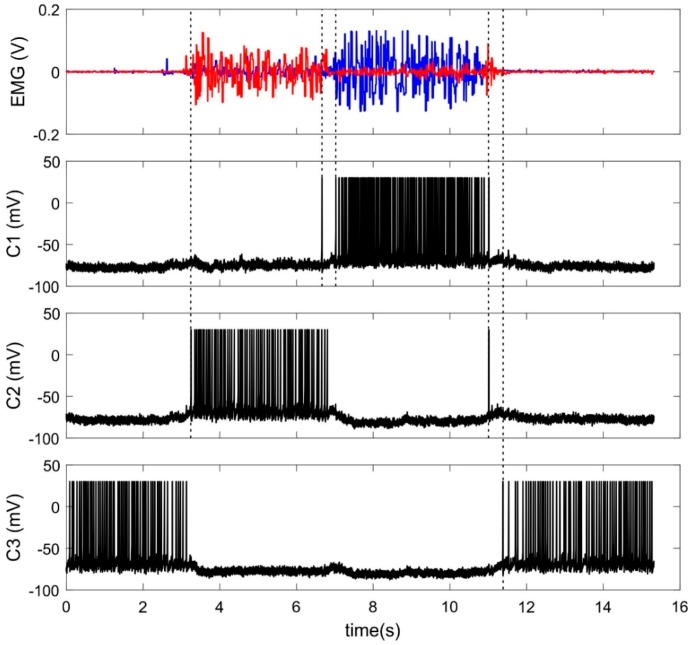
Input and output signals of SNN-classifier. Top panel: Example of two EMG channels recording muscle activity of flexor (red) and extensor (blue). C1–C3—activity (the membrane potential) of three classifier neurons.

**Figure 7 sensors-20-00500-f007:**
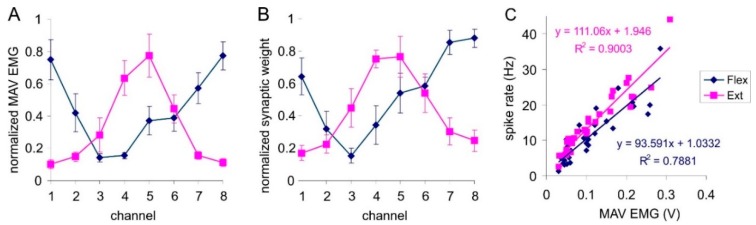
Interrelation of EMG-signal characteristics and neurons-classifiers after learning: (**A**) Average EMG-signal amplitude distribution corresponding to EMG-channel serial number; (**B**) Averaged distribution of synaptic weights corresponding to EMG-channel serial number; (**C**) Relation of spike rate to EMG-signal amplitude.

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
