# Peer review of "Competitive Learning in a Spiking Neural Network: Towards an Intelligent Pattern Classifier"

_sensors, 2020, doi:10.3390/s20020500_

Round 1
Reviewer 1 Report
The authors show that competitive STDP-based learning can learn "toy" spatiotemporal spike patterns, but not rate patterns. To learn rate patterns, an additional plasticity mechanism is needed: a "forgetting function". Next, the authors try their (unsupervised) learning algorithm on real rate-coded patterns: transduced EMG signals, and demonstrate that it can classify different muscle movements, in real time. Finally, the authors show that a supervised variation of the same learning algorithm also works here.
In my opinion, these results are new and worth sharing, but several concerns should be addressed first. 1) If I understand correctly, the EMG signal is converted into spikes using rate coding. However there is another possibility: latency coding. Each EMG patten could be converted into a wave of spikes, with the most activated units firing first. STDP is an efficient leaning mechanism in that context (see for example Masquelier & Thorpe 2007 PLoS Comp Biol) 2) The accuracy of the supervised learning algorithm was not reported! It should be reported, compared to their unsupervised algorithm, but also to other supervised learning algorithms, like backprop with surrogate gradients: Neftci et al. Surrogate Gradient Learning in Spiking Neural Networks. IEEE Signal Processing Mag This could bridge the gap between their SNN and their (backprop-trained) perceptron. Minor points: * L50: minding -> meaning * Eq 6: y -> y_j , and I think a \Sigma is missing before the dirac (multiple spikes) * Eq 9: has this rule used before? If yes, ref please. Otherwise, clearly say it is new. * L158: dished -> dashed
Reviewer 2 Report
The paper describes an attempt to use Spiking Neural Networks for classifying electromyography signals. It discusses various alternatives, including the use of temporal coding and rate coding, for unsupervised and supervised learning. An experiment is presented where EMG patterns are classified.
The paper is fairly well written, with minor typos that should be corrected. My main concern is that while the study is interesting from machine learning and neurosciences points of view, the sensors aspect is totally absent, which raises the question of the adequacy of choice of this journal to publish the work. The focus is put on the processing, and not the sensors, not even the sensors' output, barely shown Fig 5 and not even described nor commented. It would have been interesting to have an in-depth study of the features and aspects of the EMG patterns.
Other minor questions:
How exactly is the EMG signal converted to network input?
I am not sure to fully understand the results Fig 2. What is synapse order? Is it the rank of spiking? This figure, just as many others, are hard to understand and interpret since the explanations are not made explicit
Why not using the same number of neurons in the example Fig 1 and the experiments Fig 4?
In Fig 4, the colour code is not explained. What do the different hues mean?
The choice of all parameters should be justified.
Round 2
Reviewer 1 Report
All my concerns have been taken into account.
Reviewer 2 Report
Thanks for the update.
I believe that that paper can be published as it is.